# Can multi-label classification networks know what they don't know?

**Haoran Wang**[*]
Information Networking Institute
Carnegie Mellon University
haoranwa@andrew.cmu.edu

**Weitang Liu**
Department of Computer Science and Eng.
University of California, San Diego
wel022@ucsd.edu

**Alex Bocchieri**
Department of Computer Sciences
University of Wisconsin-Madison
abocchieri@wisc.edu

**Yixuan Li**[*]
Department of Computer Sciences
University of Wisconsin-Madison
sharonli@cs.wisc.edu

## Abstract

Estimating out-of-distribution (OOD) uncertainty is a major challenge for safely deploying machine learning models in the open-world environment. Improved methods for OOD detection in multi-class classification have emerged, while OOD detection methods for multi-label classification remain underexplored and use rudimentary techniques. We propose *JointEnergy*, a simple and effective method, which estimates the OOD indicator scores by aggregating label-wise energy scores from multiple labels. We show that JointEnergy can be mathematically interpreted from a joint likelihood perspective. Our results show consistent improvement over previous methods that are based on the maximum-valued scores, which fail to capture joint information from multiple labels. We demonstrate the effectiveness of our method on three common multi-label classification benchmarks, including MS-COCO, PASCAL-VOC, and NUS-WIDE. We show that JointEnergy can reduce the FPR95 by up to 10.05% compared to the previous best baseline, establishing *state-of-the-art* performance.

## 1 Introduction

Despite many breakthroughs in machine learning, formidable obstacles obstruct its deployment in the real world, where a model can encounter unknown out-of-distribution (OOD) samples. The problem of OOD detection has gained significant research attention lately [4, 18, 21, 25, 27, 28, 34, 30, 44]. OOD detection aims to identify test-time inputs that have no label intersection with training classes, thus should not be predicted by the model. Previous studies have primarily focused on detecting OOD examples in multi-class classification, where each sample is assigned to a single label. Unfortunately, this can be unrealistic in many real-world applications where images often have *multiple labels* of interest. For example, in medical imaging, multiple abnormalities may be present in a medical scan [51].

Currently, a critical research gap exists in developing and evaluating OOD detection algorithms for multi-label classification tasks that are more applicable to the real world. While one may expect solutions for multi-class setting should transfer to the multi-label setting, we show that this is far from the truth. The main challenges posed in multi-label setting stem from the need to estimate uncertainty by *jointly leveraging the information across different labels,* as opposed to relying on one

---

[*]Equal contribution. Work done while H.W was working at UW-Madison as an undergraduate researcher.

35th Conference on Neural Information Processing Systems (NeurIPS 2021).

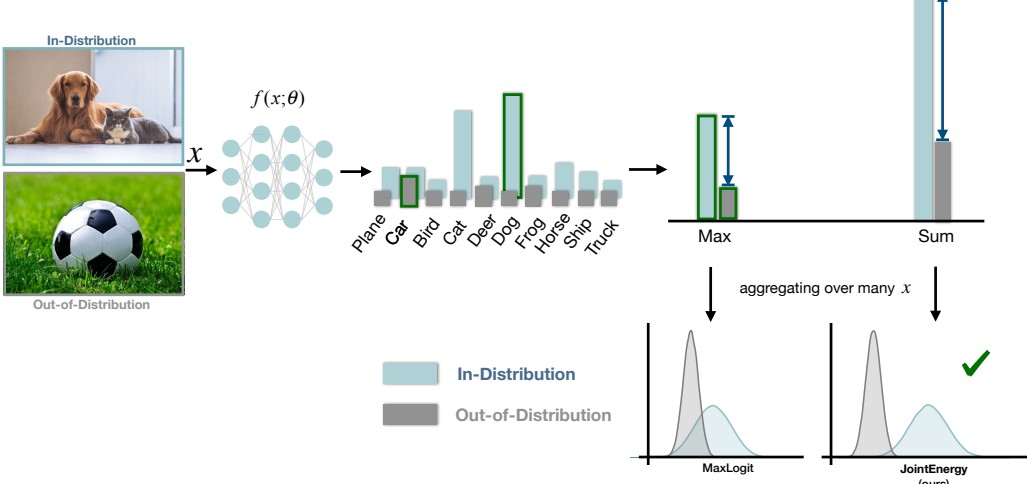

Figure 1: Out-of-distribution detection for multi-label classification networks. During inference time, input $\mathbf{x}$ is passed through classifier $f$, and label-wise scores are computed for each label. OOD indicator scores are either the maximum-valued score (denoted by green outlines) or the sum of all scores. Taking the sum results in a larger difference in scores and more separation between in-distribution and OOD inputs (denoted by red lines), resulting in better OOD detection. Plots in the bottom right depict the probability densities of MaxLogit [15] versus *JointEnergy* (ours).

dominant label. Our analysis reveals that simply using the largest model output (*i.e.*, MaxLogit) can be limiting. As a simple illustration, we contrast in Figure 1 of estimating OOD uncertainty using joint vs. single label information. MaxLogit can only capture the difference between the dominant outputs for `dog` (in-distribution) and `car` (OOD), while positive information from another dominant label `cat` (in-distribution) is dismissed.

In this paper, we address the important problem of OOD uncertainty estimation in the multi-label classification setting, and propose a simple and surprisingly effective method that jointly characterizes uncertainty from multiple labels. As a major advantage, our method circumvents the challenge to directly estimate the joint likelihood using generative models, which can be can computationally intractable to train and optimize, especially on multi-label datasets [17]. Formally, our proposed method, *JointEnergy*, derives a novel OOD score by combining label-wise energies over all labels. Despite its simplicity, we show that the JointEnergy can be theoretically interpreted from a joint likelihood perspective. The joint likelihood allows separability between in-distribution vs. OOD data, since OOD data is expected to have lower joint likelihood (*i.e.*, not associated with any of the labels). In contrast, having multiple dominant labels is indicative of an in-distribution input, which is the key aspect that JointEnergy captures. As shown in Figure 1, JointEnergy effectively amplifies the difference in scores between in-distribution and OOD inputs, compared to MaxLogit.

Extensive experiments show that JointEnergy outperforms existing methods on three common multi-label classification tasks, establishing state-of-the-art performance. For example, on a DenseNet trained with MS-COCO [29], our method reduces the false positive rate (at 95% TPR) by 10.05% when evaluated against OOD data from ImageNet [9], compared to the best performing baselines. Consistent performance improvement is observed on other multi-label tasks including PASCAL-VOC [11] and NUS-WIDE [6], as well as alternative network architecture.

Importantly, our analysis demonstrates a strong compatibility between the label-wise energy function and aggregation function, supported by both mathematical interpretation and empirical results. As an ablation, we explore the effectiveness of applying summation to popular OOD scoring functions [15, 16, 28, 27]. We find that summing labels' scores using previous methods is inferior to summing labels' energies, emphasizing the need for JointEnergy. For example, simply summing over the logits across labels results in up to 51.93% degradation in FPR95 on MS-COCO. Our study therefore underlines the importance of properly choosing both the label-wise scoring function and the aggregation method. Below we summarize our key results and contributions:

- We propose a novel method *JointEnergy*—addressing an important yet underexplored problem—OOD detection for multi-label classification networks. Our method establishes

state-of-the-art performance, reducing the average FPR95 by up to 10.05%. We show theoretical interpretation, underpinning our method from a joint likelihood perspective.

- We conduct extensive ablations which reveals important insights for multi-label OOD uncertainty estimation under (1) different aggregation functions, (2) different label-wise OOD scoring functions, and (3) the compatibility thereof.

- We curate three evaluation tasks in the multi-label setting from three real-world high-resolution image databases, which enables future research to evaluate OOD detection in a multi-label setting. Our code and dataset is released for reproducible research[2].

## 2 Background

**Multi-label Classification** Multi-label classification is the supervised learning problem where an instance may be associated with multiple labels. Let $\mathcal{X}$ (resp. $\mathcal{Y}$) be the input (resp. output) space and let $\mathcal{P}$ be a distribution over $\mathcal{X} \times \mathcal{Y}$, and let $f : \mathcal{X} \rightarrow \mathbb{R}^{|\mathcal{Y}|}$ be a neural network trained on samples drawn from $\mathcal{P}$. An input can be associated with a subset of labels in $\mathcal{Y} = \{1, 2, ..., K\}$. This set is represented by a vector $\mathbf{y} = [y_1, y_2, ..., y_K]$, where $y_i = 1$ if and only if label $i$ is associated with instance $\mathbf{x}$, and 0 otherwise. We use a convolutional neural network with shared feature space and derive the multi-label output prediction. In contrast to learning completely disjoint classifiers [47], the end-to-end training with a shared feature space is computationally more efficient than training $K$ completely independent models. This has become a de facto training mechanism for multi-label classification, with various domain applications [32, 33, 46, 51].

**Out-of-distribution Detection** The problem of OOD detection for multi-label classification is defined as follows. Denote by $\mathcal{D}_{\text{in}}$ the marginal distribution of $\mathcal{P}$ over $\mathcal{X}$, which represents the distribution of in-distribution data. At test time, the environment can incur an out-of-distribution $\mathcal{D}_{\text{out}}$ over $\mathcal{X}$. The goal of OOD detection is to define a decision function $G$ such that:

$$G(\mathbf{x}; f) = \begin{cases} 0 & \text{if } \mathbf{x} \sim \mathcal{D}_{\text{out}}, \\ 1 & \text{if } \mathbf{x} \sim \mathcal{D}_{\text{in}}. \end{cases}$$

An input is considered an OOD if it does not contain any label in the in-distribution data. In practice, $\mathcal{D}_{\text{out}}$ is often defined by a distribution that simulates anomalies encountered during deployment time, such as samples from an irrelevant distribution whose label set has no intersection with $\mathcal{Y}$ and *therefore should not be predicted by the model*.

**Energy Function** Liu *et al.* [34] first propose using free energy as a scoring function for OOD uncertainty estimation in the multi-class setting. Given a neural classifier $f(\mathbf{x}) : \mathcal{X} \rightarrow \mathbb{R}^K$ that maps an input $\mathbf{x} \in \mathcal{X}$ to K real-valued numbers as logits, a softmax function is used to derive a categorical distribution,

$$p(y_i = 1 \mid \mathbf{x}) = \frac{e^{f_{y_i}(\mathbf{x})}}{\sum_{j=1}^{K} e^{f_{y_j}(\mathbf{x})}}. \tag{1}$$

The energy model defines the probability distribution through the logits. The transformation from logits to probability distribution is by the Boltzmann distribution:

$$p(y_i = 1 \mid \mathbf{x}) = \frac{e^{-E(\mathbf{x}, y_i)}}{\int_{y'} e^{-E(\mathbf{x}, y')}} = \frac{e^{-E(\mathbf{x}, y_i)}}{e^{-E(\mathbf{x})}}.$$

Therefore, a multi-class classifier can be interpreted from an energy-based perspective by viewing the logit $f_{y_i}(\mathbf{x})$ of class $y_i$ as an energy function $E(\mathbf{x}, y_i) = -f_{y_i}(\mathbf{x})$. By equalizing the two denominators above, the *free energy* function $E(\mathbf{x})$ for any given input $\mathbf{x}$ is:

$$E(\mathbf{x}) = -\log \sum_{i=1}^{K} e^{f_{y_i}(\mathbf{x})}. \tag{2}$$

---

[2]Code and data is available: `https://github.com/deeplearning-wisc/multi-label-ood`

## 3 Method

In this work, we propose a novel method for OOD detection in multi-label classification networks, where an input can have several labels (see Figure 1). In what follows, we first introduce a label-wise energy function, and then propose *JointEnergy* that can leverage the joint information across labels for OOD uncertainty estimation.

**Label-wise Free Energy** We consider a standard pre-trained multi-label neural classifier, with a shared parameter space $\theta$ up to the penultimate feature layer. During inference time, for a given input $\mathbf{x}$, the (logit) output for the $i$-th class is:

$$f_{y_i}(\mathbf{x}) = h(\mathbf{x}; \theta) \cdot \mathbf{w}_{\text{cls}}^i, \tag{3}$$

where $\mathbf{w}_{\text{cls}}^i$ is the weight vector corresponding to class $i$, and $h(\mathbf{x}; \theta)$ is the feature vector in the penultimate layer. The predictive probability for each binary label $y_i$ is made by a binary logistic classifier:

$$p(y_i = 1 \mid \mathbf{x}) = \frac{e^{f_{y_i}(\mathbf{x})}}{1 + e^{f_{y_i}(\mathbf{x})}},$$

where $i \in \{1, 2, ..., K\}$. The logistic classifier output can be viewed as the softmax with two logits—0 and $f_{y_i}(\mathbf{x})$, respectively. For each class $y_i$, we define *label-wise free energy* as follows:

$$E_{y_i}(\mathbf{x}) = -\log(1 + e^{f_{y_i}(\mathbf{x})}), \tag{4}$$

which can be viewed as a special case of free energy in [34]. For illustration, we show the label-wise energy distribution for a subset of PASCAL-VOC classes in Figure 2 (green color). Label-wise energy captures the OOD uncertainty for a single label, but unfortunately does not capture uncertainty jointly across labels.

**JointEnergy** We propose a novel scoring function that takes into account the joint uncertainty across labels, and provide mathematical justification from a joint likelihood perspective. In particular, our method is the first to consider the joint estimation of OOD uncertainty across labels:

$$E_{\text{joint}}(\mathbf{x}) = \sum_{i=1}^{K} -E_{y_i}(\mathbf{x}) \tag{5}$$

In particular, *JointEnergy* takes the summation of label-wise energy scores across all labels. Note that in the above equations, label-wise energy $E_{y_i}(\mathbf{x})$ by definition is a negative value, and the aggregation methods output a positive value by negation. This aligns with the convention that a larger score indicates in-distribution and vice versa.

**Mathematical Interpretation** We provide mathematical interpretation for *JointEnergy*. To interpret *JointEnergy*, we first resort to the energy-based model [26], where the conditional likelihood $p(\mathbf{x} \mid y_i = 1)$ is given by:

$$p(\mathbf{x} \mid y_i = 1) = \frac{e^{-E_{y_i}(\mathbf{x})}}{\int_{\mathbf{x}|y_i} e^{-E_{y_i}(\mathbf{x})}}, \tag{6}$$

and $Z_{y_i} = \int_{\mathbf{x}|y_i} e^{-E_{y_i}(\mathbf{x})}$ is the normalized density.

We now show that *JointEnergy* can be interpreted from the joint likelihood perspective:

$$E_{\text{joint}}(\mathbf{x}) = \sum_{i=1}^{K} \log\left(p(\mathbf{x} \mid y_i = 1) \cdot Z_{y_i}\right) \tag{7}$$

$$= \sum_{i=1}^{K} \log p(\mathbf{x} \mid y_i = 1) + \underbrace{\sum_{i=1}^{K} \log Z_{y_i}}_{Z} \tag{8}$$

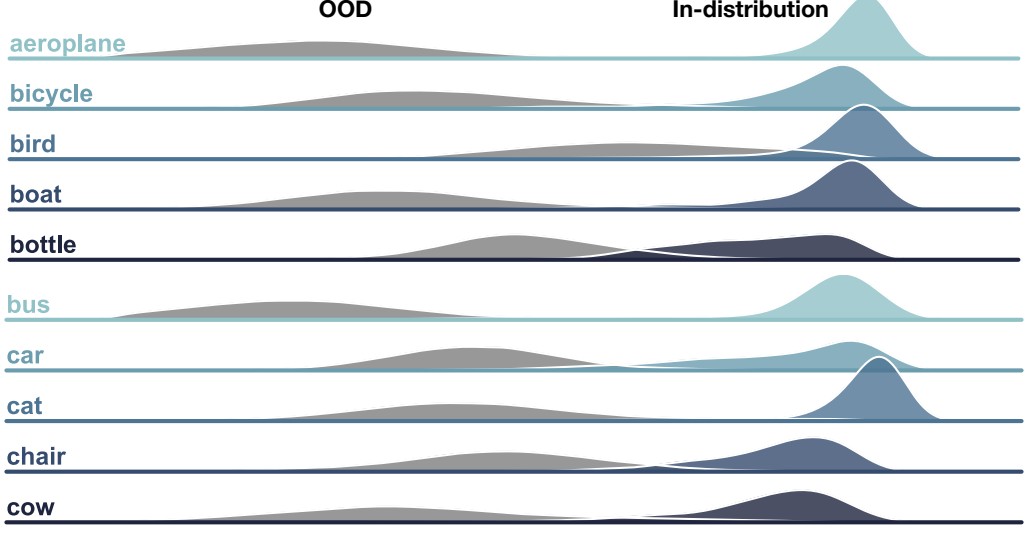

Figure 2: Label-wise energy scores $-E_{y_i}(\mathbf{x})$ distribution. The in-distribution classes (each per row) are a subset from PASCAL-VOC (green). OOD test data is from ImageNet (gray), which is the same for all labels. x-axis is in log scale for visibility.

By applying Bayesian rule for each term $\log p(\mathbf{x} \mid y_i = 1)$ in Equation 8, we have

$$E_{\text{joint}}(\mathbf{x}) = \log \prod_{i=1}^{K} \frac{p(y_i = 1 \mid \mathbf{x}) \cdot p(\mathbf{x})}{p(y_i = 1)} + Z \tag{9}$$

$$= \log \prod_{i=1}^{K} p(y_i = 1 \mid \mathbf{x}) + K \cdot \log p(\mathbf{x})$$

$$+ \underbrace{(Z - \log \prod_{i=1}^{K} p(y_i = 1))}_{C} \tag{10}$$

Given all labels $y_i$ are conditionally independent[3], we have $\prod_{i=1}^{K} p(y_i = 1 \mid \mathbf{x}) = p(y_1 = 1, y_2 = 1, ..., y_K = 1 \mid \mathbf{x})$. Therefore, Equation 10 is equivalent to:

$$E_{\text{joint}}(\mathbf{x}) = \log p(y_1 = 1, y_2 = 1, \dots, y_K = 1 \mid \mathbf{x})$$
$$+ K \cdot \log p(\mathbf{x}) + C \tag{11}$$

$$= \log \frac{p(\mathbf{x} \mid y_1 = 1, y_2 = 1, ..., y_K = 1) \cdot \prod_{i=1}^{K} p(y_i = 1)}{p(\mathbf{x})}$$
$$+ K \cdot \log p(\mathbf{x}) + C \tag{12}$$

$$= \underbrace{\log p(\mathbf{x} \mid y_1 = 1, y_2 = 1, ..., y_K = 1)}_{\text{joint conditional log likelihood, } \uparrow \text{ for in-distribution}}$$
$$+ \underbrace{(K-1) \cdot \log p(\mathbf{x})}_{\text{log data density, } \uparrow \text{ for in-distribution}} + Z \tag{13}$$

**Rationale of Equation 13** The equation above suggests that $E_{\text{joint}}(\mathbf{x})$ can be interpreted from the joint conditional log likelihood and log data density perspective. The second term is desirable for OOD detection since it reflects the underlying data density, which is higher for in-distribution data

---

[3]Note that this is sufficient but not necessary for our results to hold. Our theoretical assumption is made to ease and facilitate the interpretation from a joint likelihood perspective. Importantly, our experiments in Section 4 hold without imposing any condition and demonstrate superior performance.

Table 1: OOD detection performance comparison using JointEnergy vs. competitive baselines. We use DenseNet [19] to train on the in-distribution datasets. We use a subset of ImageNet classes as OOD test data, as described in Section 4.1. All values are percentages. ↑ indicates larger values are better, and ↓ indicates smaller values are better. **Bold** numbers are superior results. Description of baseline methods, additional evaluation results on different OOD test data, and different architecture (*e.g.*, ResNet [14]) can be found in the Appendix.

| $\mathcal{D}_{\text{in}}$ | MS-COCO | PASCAL-VOC | NUS-WIDE |
|---|---|---|---|
| | **FPR95 / AUROC / AUPR** | | |
| **OOD Score** | ↓    ↑    ↑ | | |
| MaxLogit [15] | 43.53 / 89.11 / 93.74 | 45.06 / 89.22 / 83.14 | 56.46 / 83.58 / 94.32 |
| MSP [16] | 79.90 / 73.70 / 85.37 | 74.05 / 79.32 / 72.54 | 88.50 / 60.81 / 87.00 |
| ODIN [28] | 43.53 / 89.11 / 93.74 | 45.06 / 89.22 / 83.16 | 56.46 / 83.58 / 94.32 |
| Mahalanobis [27] | 46.86 / 88.59 / 93.85 | 41.74 / 88.65 / 81.12 | 62.67 / 84.02 / 95.25 |
| LOF [3] | 80.44 / 73.95 / 86.01 | 86.34 / 69.21 / 58.93 | 85.21 / 67.75 / 89.61 |
| Isolation Forest [31] | 94.39 / 49.04 / 66.87 | 93.22 / 50.67 / 35.78 | 95.69 / 53.12 / 83.32 |
| **JointEnergy** | **33.48 / 92.70 / 96.25** | **41.01 / 91.10 / 86.33** | **48.98 / 88.30 / 96.40** |

**x**. The first term takes into account joint estimation across labels, which is new to our multi-label setting and was not previously considered in multi-class setting [34]. The first term allows even further discriminativity between in- vs. OOD data, since OOD data is expected to have lower joint conditional likelihood (*i.e.*, not associated with any of the labels). In contrast, having multiple dominant labels is indicative of an in-distribution input, which is a characteristic that *JointEnergy* captures. As a major advantage, our method circumvents the challenge to directly estimate the joint likelihood using generative models, which can be computationally intractable to train and optimize on multi-label datasets [17].

### 3.1 JointEnergy for Multi-Label OOD Detection

We propose using the JointEnergy function $E_{\text{joint}}(\mathbf{x})$ defined in Section 3 for OOD detection:

$$G(\mathbf{x}; \tau) = \begin{cases} \text{out} & \text{if } E_{\text{joint}}(\mathbf{x}) \leq \tau, \\ \text{in} & \text{if } E_{\text{joint}}(\mathbf{x}) > \tau, \end{cases} \tag{14}$$

where $\tau$ is the energy threshold, and can be chosen so that a high fraction (*e.g.*, 95%) of in-distribution data is correctly classified by $G(\mathbf{x}; \tau)$. The sensitivity analysis on $\tau$ is provided in Figure 3. A data point with higher JointEnergy $E_{\text{joint}}(\mathbf{x})$ is considered as in-distribution, and vice versa (see Fig. 1).

## 4 Experiments

In this section, we describe our experimental setup (Section 4.1) and demonstrate the effectiveness of our method on several OOD evaluation tasks (Section 4.2). We also conduct extensive ablation studies and comparative analysis that lead to an improved understanding of different methods.

### 4.1 Setup

**In-distribution Datasets** We consider three multi-label datasets: MS-COCO [29], PASCAL-VOC [11], and NUS-WIDE [6]. MS-COCO consists of 82,783 training, 40,504 validation, and 40,775 testing images with 80 common object categories. PASCAL-VOC contains 22,531 images across 20 classes. NUS-WIDE includes 269,648 images across 81 concept labels. Since NUS-WIDE has invalid and untagged images, we follow [56] and use 119,986 training images and 80,283 test images.

**Training Details** We train three multi-label classifiers, one for each dataset above. The classifiers have a DenseNet-121 backbone architecture, with a final layer that is replaced by 2 fully connected layers. Each classifier is pre-trained on ImageNet-1K and then fine-tuned with the logistic sigmoid function to its corresponding multi-label dataset. We use the Adam optimizer [23] with standard parameters ($\beta_1 = 0.9$, $\beta_2 = 0.999$). The initial learning rate is $10^{-4}$ for the fully connected layers and $10^{-5}$ for convolutional layers. We also augmented the data with random crops and random flips to obtain color images of size $256 \times 256$. After training, the mAP is 87.51% for PASCAL-VOC, 73.83% for MS-COCO, and 60.22% for NUS-WIDE. All experiments are conducted on NVIDIA GeForce RTX 2080Ti.

Table 2: Ablation study on the effect of summation for prior approaches. We use DenseNet [19] to train on the in-distribution datasets. We use ImageNet as OOD test data as described in Section 4.1. Note that *Sum* does not apply to tree-based or KNN-based approaches (e.g., LOF and Isolation Forest).

| OOD Score | $\mathcal{D}_{in}$ Aggregation | MS-COCO | PASCAL FPR95 / AUROC / AUPR ↓ ↑ ↑ | NUS-WIDE |
|---|---|---|---|---|
| Logit | Sum | 95.46 / 61.81 / 80.39 | 87.18 / 72.68 / 61.24 | 96.53 / 51.75 / 82.55 |
| Prob | Sum | 45.04 / 89.32 / 94.40 | **38.57** / 86.53 / 79.10 | 50.84 / 83.82 / 95.15 |
| ODIN | Sum | 56.56 / 84.62 / 92.24 | 50.35 / 79.45 / 70.19 | 56.26 / 81.04 / 94.34 |
| Mahalanobis | Sum | 53.43 / 87.52 / 93.35 | 44.43 / 87.76 / 79.86 | 69.05 / 80.46 / 94.09 |
| LOF | Sum | N/A | N/A | N/A |
| Isolation Forest | Sum | N/A | N/A | N/A |
| **JointEnergy (ours)** | Sum | **33.48 / 92.70 / 96.25** | 41.01 / **91.10 / 86.33** | **48.98 / 88.30 / 96.40** |

**Out-of-distribution Datasets** To evaluate the models trained on the in-distribution datasets above, we follow the same set up as in [15] and use ImageNet [9] for its generality. Besides, we evaluate against the Textures dataset [8] as OOD. For ImageNet, we use the same set of 20 classes chosen from ImageNet-22K as in [15]. These classes are chosen not to overlap with ImageNet-1k since the multi-label classifiers are pre-trained on ImageNet-1K. Specifically, we use the following classes for evaluating the MS-COCO and PASCAL-VOC pre-trained models: *dolphin, deer, bat, rhino, raccoon, octopus, giant clam, leech, venus flytrap, cherry tree, Japanese cherry blossoms, redwood, sunflower, croissant, stick cinnamon, cotton, rice, sugar cane, bamboo, and turmeric*. Since NUS-WIDE contains high-level concepts like animal, plants and flowers, we use a different set of classes that are distinct from NUS-WIDE: *asterism, battery, cave, cylinder, delta, fabric, filament, fire bell, hornet nest, kazoo, lichen, naval equipment, newspaper, paperclip, pythium, satellite, thumb, x-ray tube, yeast, zither*.

**Evaluation Metrics** We measure the following metrics that are commonly used for OOD detection: (1) the false positive rate (FPR95) of OOD examples when the true positive rate of in-distribution examples is at 95%; (2) the area under the receiver operating characteristic curve (AUROC); and (3) the area under the precision-recall curve (AUPR).

## 4.2 Results

**How does JointEnergy compare to common OOD detection methods?** In Table 1, we compare energy-based approaches against competitive OOD detection methods in literature, where *JointEnergy* demonstrates state-of-the-art performance. For fair comparisons, we consider approaches that rely on pre-trained models (without performing retraining or fine-tuning). Following the setup in [15], all the numbers reported are evaluated on ImageNet OOD test data, as described in Section 4.1. We provide additional evaluation results for the Texture OOD test dataset in the supplementary. Most baselines such as MaxLogit [15], Maximum Softmax Probability (MSP) [16], ODIN [28] and Mahalanobis [27] derive OOD indicator scores based on the maximum-valued statistics among all labels. Local Outlier Factor (LOF) [3] uses K-nearest neighbors (KNN) to estimate the local density, where OOD examples are detected from having lower density compared to

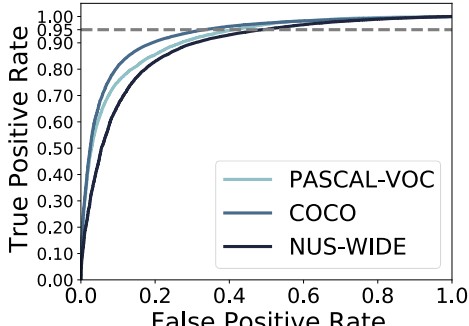

Figure 3: AUROC curves for OOD detector obtained from three in-distribution multi-label classification datasets.

their neighbors. Isolation forest [31] is a tree-based approach, which detects anomaly based on the path length from the root node to the terminating node.

Among different approaches, *JointEnergy* outperforms the best-performing baseline across all three multi-label classifiers considered. In particular, on a network trained with the MS-COCO dataset, *JointEnergy* reduces FPR95 by **10.05**%, compared to MaxLogit. We provide the AUROC curves for our method JointEnergy in Figure 3, for all three in-distribution datasets considered. The y-axis

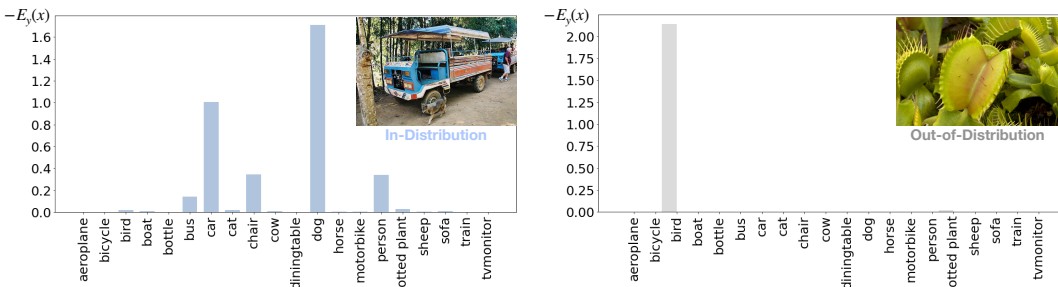

Figure 4: Label-wise energy scores $-E_{y_i}(\mathbf{x})$ for in-distribution example from PASCAL-VOC (left), and OOD input from ImageNet (right). The OOD input is misclassified using MaxLogit score since the dominant output has a high activation, making it indistinguishable from an in-distribution data's MaxLogit score. In contrast, *JointEnergy* correctly classifies both images since it results in larger differences in scores between in-distribution and OOD inputs.

is the true positive rate (TPR), whereas the x-axis is the FPR. The curves indicate how the OOD detection performance changes as we vary the threshold $\tau$ in Equation 14. We additionally evaluate on a different architecture, ResNet [14], for which we observe consistent improvement and provide details in the supplementary.

We also note here that existing approaches (such as Mahalanobis distance [27]) requires training a separate classifier for OOD detection. In contrast, JointEnergy is hyperparameter-free and easy to use in practice. In particular, the Mahalanobis approach is based on the assumption that feature representation forms class-conditional Gaussian distributions, and hence may not be well suited for the multi-label setting (which requires joint distribution to be learned).

**How do different aggregation methods affect OOD detection performance?** In Table 3, we also perform a comparative analysis of the effect of different aggregation functions that combine label-wise energy scores. As an alternative, we consider

$$E_{\max}(\mathbf{x}) = \max_i -E_{y_i}(\mathbf{x}), \tag{15}$$

which finds the largest label-wise energy score among all labels. We observe that *MaxEnergy* does not outperform *JointEnergy*, which utilizes information jointly from all the labels. The performance of *MaxEnergy* is on par with MaxLogit since *MaxEnergy*, given by $\max_i \log(1 + e^{f_{y_i}(\mathbf{x})})$, is approximately close to the MaxLogit when $f_{y_i}(\mathbf{x})$ is large. The results underline the importance of taking into account information from multiple labels, not just the maximum-valued label. This is because, in multi-label classification, the model may assign high probabilities to several classes. Theoretically, *JointEnergy* is also more meaningful, and can be interpreted from a joint likelihood perspective as shown in Section 3.

**What is the effect of applying the aggregation method to prior methods?** As an extension, we explore the effectiveness of applying the aggregation method to previous scoring functions. The results are summarized in Table 2. We calculate scores based on the logit $f_{y_i}(\mathbf{x})$, sigmoid of the logit $\frac{1}{1+e^{-f_{y_i}(\mathbf{x})}}$, ODIN score, and Mahalanobis distance score $M_{y_i}(\mathbf{x})$ for each label independently. We then perform summation across the label-wise scores as the overall OOD score. This ablation essentially replaces the *Max* aggregation with *Sum*, which helps understand the extent to which previous approaches are amenable in the multi-label setting. Note that the summation aggregation method does not apply to tree-based or KNN-based approaches such as LOF and Isolation Forest.

We found that applying summation over individual logit/MSP/ODIN/Mahalanobis scores from each label does not enhance but sometimes worsen the performance. For example, simply summing over

Table 3: Ablation study on the effect of aggregation methods: max vs summation. Values are AUROC.

| $\mathcal{D}_{\text{in}}$ | MaxEnergy | JointEnergy |
|---|---|---|
| **MS-COCO** | 89.11 | **92.70** |
| **PASCAL-VOC** | 89.22 | **91.10** |
| **NUS-WIDE** | 83.58 | **88.30** |

the logits across the labels leads to severe degradation in performance since the outputs are mixed with positive and negative numbers. On MS-COCO, the FPR degrades from 43.53% using MaxLogit to 95.46% (using SumLogit). In contrast, *JointEnergy* does not suffer from this issue. This underlines the importance of choosing proper label-wise scoring function to be compatible with the aggregation method.

**JointEnergy vs. SumProb** We highlight the advantage of JointEnergy over SumProb both empirically and theoretically. As seen in Table 2, the performance difference between JointEnergy and SumProb is substantial. In particular, on MS-COCO, our method outperforms SumProb by 11.56% (FPR95). For threshold independent metric AUROC, JointEnergy consistently outperforms SumProb by 3.38% (MS-COCO), 4.57% (PASCAL), and 4.48% (NUS-WIDE).

**Qualitative Case Study** Lastly, to provide further insights on our method, we qualitatively examine examples from the multi-label classification dataset PASCAL-VOC (in-dist.) and OOD input from the ImageNet that are correctly classified by *JointEnergy* but not MaxLogit. In Figure 4 (left), we see an in-distribution example is labeled as dog, car, chair and person, with *MaxLogit* score 1.63 and *JointEnergy* score 3.23. We also show an OOD input (Figure 4, right) with a single dominant activation on the bird class, with MaxLogit score 2.14 and *JointEnergy* score 2.19. In this example, taking the sum appropriately results in a higher score for the in-distribution image than the OOD image. Contrarily, MaxLogit score for the in-distribution image is lower than that of the OOD image, which results in ineffective detection.

# 5 Related Work

**Multi-label Classification** The task of identifying multiple classes within an input example is of significant interest in many applications [47] where deep neural networks are commonly used as the classifiers. Natural images usually contain several objects and may have many associated tags [50]. Chen *et al.* [12] used convolutional neural networks (CNN) to annotate images with 3 or 5 tags on the NUS-WIDE dataset. [5] used CNNs to tag images of road scenes from 52 possible labels. In the medical domain, Wang *et al.* [51] presented a chest X-ray dataset in which one image may contain multiple abnormalities. Multi-label classification is also prominent in natural language processing [36]. Our proposed method is therefore relevant to a wide range of applications in the real world.

**Out-of-distribution Uncertainty Estimation** Detecting and rejecting unknowns has a long history in machine learning; see [54] for a comprehensive survey of the main ideas. We highlight a few representative works in the context of deep learning. The phenomenon of neural networks' overconfidence to out-of-distribution data is revealed by Nguyen *et al.* [37]. Previous works attempt to improve the OOD uncertainty estimation by proposing the ODIN score [18, 28], Mahalanobis distance-based confidence score [27], and gradient-based GradNorm score [20]. Recent work by Liu *et al.* [34] proposed using an energy score for OOD detection, which demonstrated advantages over the softmax confidence score both empirically and theoretically. Huang and Li [21] proposed a group-based OOD detection method that scales effectively to large-scale dataset ImageNet. However, previous methods primarily focused on multi-class classification networks. In contrast, we propose a hyperparameter-free measurement that allows effective OOD detection in the underexplored *multi-label* setting, where the information from various labels is combined in a theoretically interpretable manner.

**Generative-based Out-of-distribution Detection** Generative models [10, 22, 24, 41, 45, 49] can be alternative approaches for detecting OOD examples, as they directly estimate the in-distribution density and can declare a test sample to be out-of-distribution if it lies in the low-density regions. However, as shown by Nalisnick *et al.* [35], deep generative models can assign a high likelihood to out-of-distribution data. Deep generative models can be more effective for out-of-distribution detection using likelihood ratio test [40, 43] and likelihood regret [52]. Though our work is based on discriminative classification models, we show that label-wise energy scores can be theoretically interpreted from a data density perspective. More importantly, generative based models [17] can be prohibitively challenging to train and optimize, especially on large and complex multi-label datasets that we considered (*e.g.*, MS-COCO, NUS-WIDE etc.). In contrast, our method relies on a discriminative multi-label classifier, which can be easily optimized using standard SGD.

**Energy-based Learning** Energy-based machine learning models date back to Boltzmann machines [1, 42]. Energy-based learning [26, 38, 39] provides a unified framework for many probabilistic

and non-probabilistic approaches to learning. Recent work [55] also demonstrated using energy functions to train GANs [13], where the discriminator uses energy values to differentiate between real and generated images. Xie *et al.* [53] showed that a discriminative classifier can be interpreted from an energy-based perspective. Energy-based methods are also used in structure prediction [2, 48]. Liu *et al.* [34] first proposed using energy score for OOD uncertainty estimation, which demonstrated superior performance for multi-class classification networks. In contrast, our work focuses on a multi-label setting, where we contribute both empirical and theoretical insights and demonstrate the effectiveness of utilizing information jointly from across all labels.

## 6  Conclusion and Outlook

In this work, we propose energy scores for OOD uncertainty estimation in the multi-label classification setting. We show that aggregating energies over all labels into *JointEnergy* results in better separation between in-distribution and OOD inputs compared to using information from only one label's information. Additionally, we justify the mathematical interpretation of *JointEnergy* from a joint likelihood perspective. *JointEnergy* obtains better OOD detection performance compared to competitive baseline methods, establishing new state-of-the-art on this task. Applications of multi-label classification can benefit from our methods, and we anticipate further research in OOD detection to extend this work to tasks beyond image recognition. We hope our work will increase the attention toward a broader view of OOD uncertainty estimation for multi-label classification.

## 7  Societal Impact

Our project aims to improve the dependability and trustworthiness of modern machine learning models for multi-label classification. This stands to benefit a wide range of fields and societal activities. We believe out-of-distribution uncertainty estimation is an increasingly critical component of systems that range from consumer and business applications (e.g., digital content understanding) to transportation (e.g., driver assistance systems and autonomous vehicles), and to health care (e.g., unseen disease identification). Many of these applications require multi-label classification models in operation. Through this work and by releasing our code, we hope to provide machine learning researchers a new methodological perspective and offer machine learning practitioners an easy-to-use tool that renders safety against OOD data in the real world. While we do not anticipate any negative consequences to our work, we hope to continue to build on our framework in future work.

## Acknowledgement

Research is supported by the Office of the Vice Chancellor for Research and Graduate Education (OVCRGE) with funding from the Wisconsin Alumni Research Foundation (WARF).

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
