# Appendix

## A  Evaluation on different architecture

We provide additional evaluation results for ResNet [14]. The classifiers have a ResNet-101 backbone architecture, but with a final layer that is replaced by 2 fully connected layers. Each classifier is pre-trained on ImageNet-1K and then fine-tuned with the logistic sigmoid function to its corresponding multi-label dataset. We use the same training settings as in the main paper. After training, the mAP is 87.73% for PASCAL-VOC, 72.77% for MS-COCO, and 61.47% for NUS-WIDE.

In Table 4, we show the performance comparison of various OOD detection approaches, evaluated on ImageNet as the OOD test set. The ablation of applying summation over baseline methods is provided in Table 5.

Table 4: OOD detection performance comparison using energy-based approaches vs. competitive baselines. We use ResNet [14] to train on the in-distribution datasets. We use a subset of ImageNet classes as OOD test data, as described in Section 4.1. All values are percentages. ↑ indicates larger values are better, and ↓ indicates smaller values are better. **Bold** numbers are superior results.

| $\mathcal{D}_{in}$ | MS-COCO | PASCAL-VOC | NUS-WIDE |
|---|---|---|---|
| | | FPR95 / AUROC / AUPR | |
| **OOD Score** | | ↓      ↑      ↑ | |
| MaxLogit [15] | 34.54 / 90.93 / 94.30 | 36.32 / 91.04 / 82.68 | 58.05 / 83.07 / 94.21 |
| MSP [16] | 77.92 / 72.43 / 84.34 | 69.85 / 78.24 / 67.93 | 88.75 / 59.19 / 86.40 |
| ODIN [27] | 34.58 / 90.26 / 93.69 | 36.32 / 91.04 / 82.68 | 58.05 / 83.07 / 94.21 |
| Mahalanobis [26] | 94.04 / 49.49 / 70.71 | 78.02 / 70.93 / 59.84 | 61.33 / 83.75 / 95.15 |
| LOF [3] | 74.30 / 74.87 / 85.82 | 76.71 / 67.54 / 55.35 | 85.42 / 69.37 / 90.36 |
| Isolation Forest [30] | 99.06 / 37.59 / 63.43 | 98.64 / 41.94 / 33.50 | 96.59 / 50.75 / 82.91 |
| **JointEnergy** (ours) | **31.51 / 92.68 / 96.15** | **31.96 / 92.32 / 86.87** | **50.25 / 88.12 / 96.34** |

Table 5: Ablation study on the effect of aggregation methods for prior approaches. We use ResNet [14] to train on the in-distribution datasets. We use ImageNet as OOD test data as described in Section 4.1. Note that *Sum* is not applicable to tree-based or KNN-based approaches (*e.g.*, LOF and Isolation Forest).

| OOD Score | Aggregation | $\mathcal{D}_{in}$ MS-COCO | PASCAL | NUS-WIDE |
|---|---|---|---|---|
| | | FPR95 / AUROC / AUPR | | |
| | | ↓      ↑      ↑ | | |
| **Logit** | Sum | 95.63 / 53.52 / 73.25 | 96.36 / 49.44 / 43.07 | 96.49 / 49.83 / 81.78 |
| **Prob** | Sum | 43.69 / 87.21 / 93.14 | 35.97 / 84.68 / 76.61 | 55.86 / 82.97 / 94.92 |
| **ODIN** | Sum | 43.69 / 87.21 / 93.14 | 53.77 / 74.50 / 67.15 | 55.24 / 81.84 / 94.59 |
| **Mahalanobis** | Sum | 94.47 / 46.82 / 67.06 | 78.56 / 70.84 / 59.34 | 62.79 / 83.19 / 94.96 |
| **LOF** | Sum | N/A | N/A | N/A |
| **Isolation Forest** | Sum | N/A | N/A | N/A |
| **Energy** | Sum | **31.51 / 92.68 / 96.15** | **31.96 / 92.32 / 86.87** | **50.25 / 88.12 / 96.34** |

## B  Evaluation on different OOD test data

In addition to ImageNet, we also evaluate on a different OOD test dataset, Textures [7]. The results are reported in Table 6 and Table 7.

## C  Baseline Methods

In multi-label classification, the prediction for each label $y_i$ with $i \in \{1, 2, ..., K\}$ is independently made by a binary logistic classifier:

$$p(y_i \mid \mathbf{x}) = \frac{e^{f_{y_i}(\mathbf{x})}}{1 + e^{f_{y_i}(\mathbf{x})}}.$$

Table 6: Texture as OOD data. We use ResNet [14] to train on the in-distribution datasets. All values are percentages. ↑ indicates larger values are better, and ↓ indicates smaller values are better. **Bold** numbers are superior results.

| $\mathcal{D}_{\text{in}}$ | MS-COCO | PASCAL | NUS-WIDE |
|---|---|---|---|
| | | **FPR95 / AUROC / AUPR** | |
| **OOD Score** | ↓ | ↑ ↑ | |
| MaxLogit [15] | 14.63 / 96.10 / 99.32 | 12.36 / 96.22 / 96.97 | 38.46 / 87.42 / 97.19 |
| MSP [16] | 60.82 / 83.70 / 97.05 | 41.81 / 89.76 / 93.00 | 83.09 / 63.41 / 92.48 |
| ODIN [27] | **12.22** / 96.18 / 99.29 | 12.36 / 96.22 / 96.97 | 38.46 / 87.42 / 97.19 |
| Mahalanobis [26] | 44.61 / 85.71 / 97.41 | 19.17 / 96.23 / **97.90** | 36.19 / 91.36 / 98.52 |
| LOF [3] | 70.16 / 74.73 / 94.96 | 89.49 / 60.37 / 76.70 | 64.27 / 78.23 / 95.94 |
| Isolation Forest[30] | 95.55 / 53.21 / 90.45 | 99.59 / 20.89 / 50.11 | 93.07 / 51.01 / 89.17 |
| **JointEnergy** (ours) | 12.82 / **96.84 / 99.54** | **10.87 / 96.78** / 97.87 | **31.68 / 92.43 / 98.65** |

Table 7: Ablation study on the effect of aggregation methods for prior approaches. We use ResNet [14] to train on the in-distribution datasets. We use Texture [7] as OOD test data as described in Section 4.1. Note that *Sum* is not applicable to tree-based or KNN-based approaches (*e.g.*, LOF and Isolation Forest).

| | $\mathcal{D}_{\text{in}}$ | MS-COCO | PASCAL | NUS-WIDE |
|---|---|---|---|---|
| | | | **FPR95 / AUROC / AUPR** | |
| **OOD Score** | **Aggregation** | | ↓ ↑ ↑ | |
| **Logit** | Sum | 95.63 / 53.52 / 73.25 | 96.36 / 49.44 / 43.07 | 92.38 / 52.72 / 89.21 |
| **Prob** | Sum | 43.69 / 87.21 / 93.14 | 35.97 / 84.68 / 76.61 | 34.88 / 90.76 / 98.57 |
| **ODIN** | Sum | 43.69 / 87.21 / 93.14 | 53.77 / 74.50 / 67.15 | 35.27 / 89.36 / 98.31 |
| **Mahalanobis** | Sum | 45.62 / 84.34 / 97.02 | 19.45 / 96.09 / 97.80 | 37.55 / 91.04 / 98.47 |
| **LOF** | Sum | N/A | N/A | N/A |
| **Isolation Forest** | Sum | N/A | N/A | N/A |
| **Energy** | Sum | **12.82 / 96.84 / 99.54** | **10.87 / 96.78 / 97.87** | **31.68 / 92.43 / 98.65** |

We consider the following baselines methods under *maximum* aggregation:

$$\textbf{MaxLogit} = \max_i f_{y_i}(\mathbf{x}) \tag{16}$$

$$\textbf{MSP} = \max_i \frac{e^{f_{y_i}(\mathbf{x})}}{\sum_j^K e^{f_{y_j}(\mathbf{x})}} \tag{17}$$

$$\textbf{ODIN} = \max_i \frac{e^{f_{y_i}(\hat{\mathbf{x}})/T}}{1 + e^{f_{y_i}(\hat{\mathbf{x}})/T}} \tag{18}$$

$$\textbf{Mahalanobis} = \max_i -(\phi(\hat{\mathbf{x}}) - \hat{\mu}_{y_i})^{\mathsf{T}} \hat{\Sigma}^{-1} (\phi(\hat{\mathbf{x}}) - \hat{\mu}_{y_i}) \tag{19}$$

In particular, ODIN was originally designed for multi-class but we adapt for the multi-label case by taking the maximum of calibrated label-wise predictions. The input perturbation is calculated using $\hat{\mathbf{x}} = \mathbf{x} - \epsilon \text{sign}(-\nabla \ell_{\hat{y}_i})$, where $\ell_{\hat{y}_i}$ is the binary cross-entropy loss for the label $\hat{y}_i$ with the largest output, i.e., $\hat{y}_i = \arg\max_i p(y_i = 1 \mid \mathbf{x})$. For Mahalanobis distance, we extract the feature embedding $\phi(\mathbf{x})$ for a given sample. $\hat{\mu}_{y_i}$ is the class conditional mean for label $y_i$, and $\hat{\Sigma}^{-1}$ is the covariant matrix.

### C.1 Validation data for baselines

We use a combination of the following validation datasets to select hyperparameters for ODIN [27] and Mahalanobis [26]. The validation set consists of:

- Gaussian noise sampled i.i.d. from an isotropic Gaussian distribution;
- uniform noise where each pixel is sampled from $U = [-1, 1]$;
- In-distribution data corrupted into OOD data by applying (1) pixel-wise arithmetic mean of random pair of in-distribution images; (2) geometric mean of random pair of in-distribution images; and (3) randomly permuting 16 equally sized patches of an in-distribution image.

## C.2 Hyperparameter tuning for baselines

ODIN [27] and Mahalanobis [26] require hyper-parameter tuning, such as temperature and magnitude of noise $\epsilon$. We use the validation data above for selecting the optimal hyperparameters. For ODIN, temperature T is chosen from [1,10,100,1000] and the perturbation magnitude $\epsilon$ is chosen from 21 evenly spaced numbers starting from 0 and ending at 0.004. For Mahalanobis, the perturbation magnitude $\epsilon$ is chosen from [0, 0.0005, 0.0014, 0.001, 0.002, 0.005]. The optimal parameters are chosen to minimize the FPR at TPR95 on the validation set.

# D  Ablation Study: JointEnergy with Top Labels

Our method can generalize to the case when using only the top-$k$ predictions in the extreme multi-label classification case, *i.e.*, when the number of labels is very large. This in theory holds true when the some labels' outputs are relatively small especially when the logits are negative, and hence the label-wise energy $E_{y_i} = -\log(1 + e^{f_{y_i}(\mathbf{x})}) \approx 0$. In this case, omitting these labels does not largely affect the overall JointEnergy score and detection performance. The results are shown in Table 8.

Table 8: Ablation study on JointEnergy with top-$k$ labels. $k$ is the average number of labels of a training image (estimated empirically on the entire training population) for each dataset. Specifically, $k$ is 3 for MS-COCO, 2 for PASCAL-VOC, and 2 for NUS-WIDE. Dataset and model settings are the same as in Table 1.

| $\mathcal{D}_{\text{in}}$ | MS-COCO | PASCAL | NUS-WIDE |
|---|---|---|---|
| | | **FPR95 / AUROC / AUPR** | |
| **OOD Score** | | ↓    ↑    ↑ | |
| JointEnergy (all) | **33.48 / 92.70 / 96.25** | **41.01 / 91.10 / 86.33** | **48.98 / 88.30 / 96.40** |
| JointEnergy (top-$k$) | 37.85 / 91.69 / 95.71 | 43.63 / 90.39 / 85.35 | 53.00 / 86.35 / 95.68 |
| JointEnergy (top-5) | 35.83 / 92.19 / 95.99 | 41.36 / 91.00 / 86.22 | 49.87 / 87.94 / 96.28 |
| JointEnergy (top-10) | 34.43 / 92.52 / 96.16 | 41.02 / 91.09 / 86.32 | 49.35 / 88.22 / 96.37 |
| JointEnergy (top-20) | 33.72 / 92.66 / 96.23 | 41.01 / 91.10 / 86.33 | 49.14 / 88.28 / 96.39 |
| JointEnergy (top-40) | 33.53 / 92.70 / 96.25 | - / - / - | 48.98 / 88.30 / 96.40 |
| JointEnergy (top-60) | 33.48 / 92.70 / 96.25 | - / - / - | 49.03 / 88.30 / 96.40 |