# OpenReview forum: "Can multi-label classification networks know what they don’t know?"
_NeurIPS.cc/2021/Conference — NeurIPS 2021 Poster_

### Official Review · Reviewer_ggDB · 2021-06-30

**Rating:** 7
**Confidence:** 3

**Summary:**

The paper proposes a new method for OOD detection in the multi-label setting, called JointEnergy.
The authors claim that JointEnergy can be mathematically interpreted from a joint likelihood perspective, and that their model achieves sota results on a variety of benchmarks.

**Limitations And Societal Impact:**

Yes

**Main Review:**

**Significance**: high - the problem of OOD detection is well studied for multi-class classification problems, but there is little work on OOD detection for multi-label classification problems.

**Clarity**: low - the authors do not provide any background on energy-based models, which is necessary to follow the theoretical analysis.

**Quality**: The experimental analysis shows that the proposed solution is very effective, however I have some concerns with the theoretical analysis.

1. the authors define the label-wise free energy as: $E_{y_i}({x}) = - \log(1 + \exp^{f_{y_i (x)}})$. However, they do not explain neither  why they defined the energy function in that way nor which properties the function has that make it an energy function.
This does not allow the reviewers to check whether such definition (and hence all the subsequent calculations) are correct.
I would advise the authors to look into how [1] defined the energy function in the case considered, as it is very clear and easy to follow.

2. the authors make the strong assumption of label independence. However, if we assume the label independence (which is almost never verified in practice), then we are basically dealing with a series of binary classification problems. The work would be more complete if this assumption was dropped.

**Novelty**: medium - the paper presents a novel and simple idea that is able to beat sota models on the task of OOD detection.
However, the strong label independence assumption hinders its novelty.

[1] Weitang Liu, Xiaoyun Wang, John Owens, and Yixuan Li. Energy-based out-of-distribution detection. Advances in Neural Information Processing Systems, 2020.

**After author response**: the authors have addressed my concerns, thus I change my score from 6 to 7. Congrats!


**Time Spent Reviewing:**

3

---

> ### Author Response · Authors · 2021-08-10
> **Thank you for the positive feedback!**
>
> We thank the reviewer for the insightful and constructive feedback, which helped us improve the draft. We are glad the reviewer found our method novel, significant, and well-studied.
>
> **1. Clarification on the definition of energy function**
>  Excellent suggestion! We agree with the reviewer that more context on the definition would facilitate understanding our method and derivation. To provide more details, the sigmoid function $\frac{e^{f_{y_i}(\mathbf{x})}}{1+e^{f_{y_i(\mathbf{x})}}}$ can be viewed as a special case of softmax with two logits: 0 and $f_{y_i}(\mathbf{x})$. In [1], the authors have shown the connection between free energy interpretation and softmax. A softmax function is used to derive a categorical distribution,
> $$
>     p(y_i \mid \mathbf{x}) = \frac{e^{f_{y_i}(\mathbf{x})}}{\sum_{j=1}^K e^{f_{y_j}(\mathbf{x})}},
> $$
> which indicates the probability for an input $\mathbf{x}$ to be of class $y_i$, with $i \in \{1,2,...,K\}$.
>
> On the other hand, the energy model defines the probability distribution through the logits. The translation from logits to probability distribution is by the Boltzmann distribution:
> $$
>     p(y_i \mid \mathbf{x}) = \frac{e^{-E(\mathbf{x},y_i)}}{\sum_{j=1}^K e^{-E(\mathbf{x},y_j)}} = \frac{e^{-E(\mathbf{x},y_i)}}{e^{-E(\mathbf{x})}},
> $$
> where the logit $f_{y_i}(\mathbf{x})$ of class $y_i$ can be viewed as an energy function $E(\mathbf{x},y_i) = -f_{y_i}(\mathbf{x})$. By equalizing the two denominators above, the free energy function $E(\mathbf{x})$ for any given input $\mathbf{x}$ is:
> $$
>   E(\mathbf{x})=- \text{log}\sum_{i=1}^K e^{f_{y_i}(\mathbf{x})}.
> $$
>
> As a reduction of softmax, the label-wise free energy has the form of $E_{y_i} = -\log (1+e^{f_{y_i}})$. We have revised our draft and incorporated a more expansive discussion on the definition. Thank you again for pointing this out!
>
> **2. Clarification on the assumption**
>  Great question! In particular, all of our empirical results are based on convolutional neural networks with a shared feature space that captures label dependencies, which is different from training K completely disjoint classifiers. This is a common training practice in use today for multi-label classification, and has been long established (e.g., see Read et al. 2014 in settings of deep belief networks). Our theoretical justification is based on the conditional label independence assumption, which facilitates the interpretation from a joint likelihood perspective. Note that this is **sufficient but not necessary** for our results to hold. Importantly, our experiments in Section 4 hold without imposing any conditions and demonstrate state-of-the-art performance. This is discussed in footnote 1 but we have highlighted it for better clarity.
>
> [1] Weitang Liu, Xiaoyun Wang, John Owens, and Yixuan Li. Energy-based out-of-distribution detection. Advances in Neural Information Processing Systems, 2020.

---

### Official Review · Reviewer_p44K · 2021-07-07

**Rating:** 7
**Confidence:** 3

**Summary:**

The paper presents a new score - JointEnergy, as a sum of free energy across labels to detect out of distribution inputs in a multi-label classification setting. It presents the definition of this new score, motivates it theoretically by connecting it to joint likelihood of data, and demonstrates the superior performance of this score over existing alternatives on 3 different datasets. The results are interesting and significant.

**Ethical Concerns:**

I do not anticipate any significant ethical concerns of this work.

**Ethics Review Area:**

["I don’t know"]

**Limitations And Societal Impact:**

It is not clear how the new score performs against baselines in detecting OOD inputs in other tasks (other than vision - like language, speech/audio, structured datasets etc.). It is also not clear how it performs when training a different model for each label (i.e without sharing a pre-trained representation). The authors can better address these 2 aspects through both theoretical analysis and experiments.

I do not anticipate any significant negative societal impact of this work.

**Main Review:**

Originality: The problem of OOD detection in multi-label classification setting is under studied. This paper extends existing ideas in a novel way to achieve superior detection performance.

Quality: While the proposed new OOD  detection score is simple and straightforward extension of the free energy interpretation of logit scores, the new score is better motivated theoretically, and shows superior performance to existing alternatives. It uses more information in the scores of all labels instead of just the dominant label and hence is better able to detect OOD inputs. Despite its simplicity, the new score has better performance. It is also free of hyper parameters, which makes it easier to implement. The method applies to general multi-label classification and hence should be extensible to tasks beyond vision (although the paper does not demonstrate it).

Clarity: The paper is written well. The new score is motivated well, is theoretically justified, and experiments are simple and conclusive

Significance: Despite its simplicity, the new score seems to be quite powerful in detecting OOD inputs in multi-label classification, and can become a significant evaluation metric for this purpose in the future.


**Time Spent Reviewing:**

2.5 hours

---

> ### Author Response · Authors · 2021-08-10
> **Thank you for the insightful comments!**
>
> We are more encouraged by the positive comments from the reviewer, which raise insightful points on not only improving our draft but also extending our work in future research. We are glad the reviewer found our method novel, simple, straightforward, achieving superior performance, and a significant contribution to the field.
>
> **1. Other modalities**
>  As the reviewer recognizes, our method is generally applicable for multi-label classification and hence has the potential for tasks beyond vision. We focused on image modality as it’s directly motivated by the prevalence in image datasets where multiple objects could occur. Moreover, OOD detection literature in the multi-class setting has also been commonly evaluated on the image datasets, hence we thought it’d be a natural modality to explore. As the first work to formally establish the multi-label OOD detection problem, we believe it’s much easier for the community to build on our work if we start with a more focused domain, understand it well and thoroughly, before going breadth. With that being said, we certainly agree with the importance of extending our work beyond image datasets. We believe it can be a worthy follow-up work where one can delve much deeper into the text/speech domains, carefully construct the evaluations and thoroughly compare the performance. We have added a paragraph acknowledging this limitation and potential future direction. Thank you again for the suggestion!
>
> **2. Training each label independently**
>  Our training is based on a common practice in use today for multi-label classification. Our method and theory would still hold and translate to the settings where each label is trained independently. In other words, derivation in Equation (4)-Equation (11) would still be valid. For example, in Equation (4), you can think of the normalized density now depends on an independent network parameterization without sharing---this does not affect the conclusion as Z is ultimately a constant for all inputs $\mathbf{x}$. Parameter and feature sharing can allow better multi-label classification accuracy on the ID task and computational efficiency, especially when labels have common low-level features. We have added a discussion on this in our updated draft.

---

### Official Review · Reviewer_GeqC · 2021-07-16

**Rating:** 7
**Confidence:** 4

**Summary:**

This paper studies a new task that detects out-of-distribution (OOD) data in the multi-label setting. Inspired by recent progress in multi-class classification problems, the authors proposed a joint energy score to achieve their goal. Specifically, the proposed method sums up the label-wise energy score. Then, it is used as a criterion to determine which samples are out-of-distribution by thresholding. Empirical results demonstrate that the proposed method achieved promising results on COCO, NUSWIDE, and VOC benchmark datasets.

**Limitations And Societal Impact:**

I have mixed thoughts on this paper. On one hand, the proposed joint density score is a simple generalization from the previous work. On the other hand, I believe this work can inspire a series of works and appreciate its contribution to well-defining new research topics and establishing a simple yet effective benchmark.

Moreover, I recommend the authors further polished the mathematical interpretation section.

**Main Review:**

Pros:
1. While the out-of-distribution detection task has been widely studied in the multi-class classification context, it could be very tricky in multi-label learning, because there can be combinational objects and some combinations may never occur.
2. The experimental results are thorough to verify the effectiveness of the proposed method, which I believe will be beneficial to the community.

Cons:
1. The definition of energy score in [1] is an unconditional version. However, the authors explain their criterion through the line of the conditional version of energy. They should refer their definition to other works that give the conditional form of energy, e.g. https://arxiv.org/pdf/2002.11537.pdf.
2. The authors claim that in the case of OOD, each of the conditional likelihoods should be small, hence the product. Actually, we are considering a conditional probability $P(X=x|Y_1=1,Y_2=1,...,Y_k=1)$. I suppose the only property of OOD data is: given an OOD data, $P(y_1=0,...,y_K=0|X=x)$ is far larger than other data. It is not an obvious induction to conclude $P(X=x_1|Y_1=1,Y_2=1,...,Y_k=1)<P(X=x_2|Y_1=1,Y_2=1,...,Y_k=1)$, s.t. $x_1$ is OOD while $x_2$ is in-distribution. Actually, we can find the key in Eq. (8) where the production of per-label posterior is low for in-distribution data. Thus, we needn't further explore the conditional density.

[1] LeCun Y, Chopra S, Hadsell R, et al. A tutorial on energy-based learning[J]. Predicting structured data, 2006, 1(0).

**Time Spent Reviewing:**

2

---

> ### Author Response · Authors · 2021-08-10
> **Thank you for the constructive feedback!**
>
> We thank the reviewer for finding our method simple and effective, well-defining a new research problem, and potentially inspiring a series of works. We also appreciate the feedback regarding math clarity, which we address below.
>
> **1. Definition of the conditional form of energy**
>  Thanks for pointing that out! We agree and have revised our draft with a clear definition of conditional energy, and referred to the work shared.
>
> **2. Clarification on the mathematical interpretation**
>  As the reviewer pointed out, the OOD detection criterion should ideally be based on the conditional likelihood where $$
> P(X=x_1 |Y_1=1, Y_2=1,...,Y_k=1) < P(X=x_2 |Y_1=1, Y_2=1,...,Y_k=1).
> $$
> Here $x_2$ is an ID sample and $x_1$ is an OOD sample. However, estimating this joint likelihood directly can be intractable, especially in multi-label settings as it requires sampling combinatorially in the high-dimensional space (see **L37-39**). Our theoretical contribution showed how this can be conveniently captured by JointEnergy.
>
> To see why the above equation holds, we can use Bayes' rule by expressing it in terms of posterior and prior.
> $$
> P(X=x_1 |Y_1=1, Y_2=1,...,Y_k=1) = \frac{P(Y_1=1, Y_2=1,...,Y_k=1|x_1) \cdot P(x_1)}{P(Y_1=1, Y_2=1,...,Y_k=1)}
> $$
>
> $$= \frac{\prod_{i=1}^k P(Y_i=1|x_1) \cdot P(x_1)}{P(Y_1=1, Y_2=1,...,Y_k=1)}$$
>
> The numerator has two terms that are both $\downarrow$ for OOD: the product of per-label posterior probability (which is small for OOD as OOD data is not associated with any label), and the probability density p(x) which is also small for OOD. So overall, the conditional likelihood is also smaller for OOD than ID data. As you pointed out, this is also reflected in our JointEnergy in Equation (8). Likelihood and posterior have similar trends, but the former has a direct mathematical interpretation from a data density perspective. We have revised our draft to make this connection clearer. As suggested, we have also polished the mathematical derivation (including making the notation clear by using $Y_i=1$ explicitly in multiple missing places). Thank you again for the careful reading!

---

### Official Review · Reviewer_QXpR · 2021-07-21

**Rating:** 6
**Confidence:** 4

**Summary:**

This paper studies the out-of-distribution detection problem in the multi-label classification setting. The authors propose a scoring function named JointEnergy to measure the prediction uncertainty of a given instance. In particular, the JointEnergy aggregates the label wise free-engery by summation and a hard thresholding on the value of JointEnergy is used to determine whether a given input is OOD or not. It is shown that the propose scoring function can be decomposed into three terms: joint conditional log-likelihood on all classes, un-conditional log data density and a constant term. Empirical results on three multi-label classification datasets show that the proposed method can achieve lower false positive rate (FPR95) while having higher area under curve scores (AUROC, AUPR).

**Limitations And Societal Impact:**

This paper studies the out-of-distribution detection problem in the multi-label classification setting. To my understanding, the work poses no negative societal impact.

**Main Review:**

The scoring function takes a seemingly trivial format which is the simple summation of the per-label free energy for a given input. However, the authors justify the choice by associate the JointEnergy with the joint conditional log-likelihood and the log data density, both are expected to be high for in-distribution samples. The experiments also shows that the choice is valid. In general, I think this paper proposes a simple but effective method to handle an often overlooked problem and vote for accept. Several minor comments:

(1)  The label-wise free energy form is based on that the predictive probability is given by sigmoid(f(x)). For the experiments, models are fine-tuned with the logistic sigmoid functions. What's the connection between how the model is trained and how the JointEnergy estimator would work? Can the JointEnergy estimator work on models trained with other logit smoothing functions?

(2) The two terms, log data density and joint conditional log-likelihood, in the JointEnergy seems to be scaling linearly with the number of labels K. Therefore the contributions of the two terms will remain roughly the same even with very large K. In the case where K is very large, i.e. extreme multi-label classification, only top-k predictions are given. I wonder if the JointEnergy can be generalized to that case where the logit from k+1 to K are truncated.

(3) It would be helpful to do some ablation study on how the log data density and log-likelihood would perform on their own.

**Time Spent Reviewing:**

3h

---

> ### Author Response · Authors · 2021-08-10
> **Thank you for the helpful comments!**
>
> We are glad the reviewer found our method simple and effective, studying an overlooked problem, with proper justification both in theory and experiments. We appreciate the detailed comments and suggestions, which have helped us improve our draft. We address the comments raised by the reviewer below:
>
> **1. The connection between how the model is trained and JointEnergy estimator**
>  The form of JointEnergy estimator works aligns with how the model is trained. As the reviewer pointed out, our model is trained on the logistic function, which is a common training objective in use today for multi-label classification. We use the training scheme for its simplicity, generality as well as inherent connection with the JointEnergy estimator. The sigmoid function $\frac{e^{f_{y_i}(\mathbf{x})}}{1+e^{f_{y_i(\mathbf{x})}}}$ can be viewed as a special case of softmax with two logits: 0 and $f_{y_i}(\mathbf{x})$. In [1], the authors have shown the connection between free energy interpretation and softmax. To provide detailed context, a softmax function is used to derive a categorical distribution,
> $$
>     p(y_i \mid \mathbf{x}) = \frac{e^{f_{y_i}(\mathbf{x})}}{\sum_{j=1}^K e^{f_{y_j}(\mathbf{x})}},
> $$
> which indicates the probability for an input $\mathbf{x}$ to be of class $y_i$, with $i \in \{1,2,...,K\}$.
>
> On the other hand, the energy model defines the probability distribution through the logits. The translation from logits to probability distribution is by the Boltzmann distribution:
> $$
>     p(y_i \mid \mathbf{x}) = \frac{e^{-E(\mathbf{x},y_i)}}{\sum_{j=1}^K e^{-E(\mathbf{x},y_j)}} = \frac{e^{-E(\mathbf{x},y_i)}}{e^{-E(\mathbf{x})}},
> $$
> where the logit $f_{y_i}(\mathbf{x})$ of class $y_i$ can be viewed as an energy function $E(\mathbf{x},y_i) = -f_{y_i}(\mathbf{x})$. By equalizing the two denominators above, the free energy function $E(\mathbf{x})$ for any given input $\mathbf{x}\in \mathbb{R}^D$ is:
> $$
>   E(\mathbf{x})=- \text{log}\sum_{i=1}^K e^{f_{y_i}(\mathbf{x})}.
> $$
>
> Therefore, there is an inherent connection between softmax and free energy. As a reduction of softmax, the use of label-wise energy has an inherent connection with the sigmoid function. If the model is trained with other logit smooth functions, the form of JointEnergy should change accordingly.
>
>  **2. JointEnergy using top-k prediction**
>  Excellent question! We reckon with the reviewer that our method can generalize to the case when taking the top-k predictions in the extreme multi-label classification case. This in theory should hold true when the truncated logits are relatively small especially when the logits are negative, and hence the label-wise energy $E_{y_i} = -\log (1+e^{f_{y_i}(\mathbf{x})}) \approx 0$. In this case, whether taking into account the truncated logit does not largely affect the overall JointEnergy score and detection performance.
> To verify this idea, we choose k for each in-distribution dataset, based on the average number of labels of a training image (estimated empirically on the entire training population). Specifically, k is 3 for MS-COCO, 2 for PASCAL-VOC, and 2 for NUS-WIDE. Dataset and model settings are the same as in Table 1. The performance for each ID data is summarized in the table below for your reference. As you hypothesized, our method can generalize to top-k predictions (and the performance is going to be increasingly similar as we make k even larger).
>
> (note: we use the format of FPR95 / AUROC / AUPR in each entry below)
>
>
> | $\mathcal{D}_{\text{in}}$ | MS-COCO | PASCAL-VOC | NUS-WIDE |
> |:-------------------:|:---------------------:|:---------------------:|:---------------------:|
> | JointEnergy | 33.48 / 92.70 / 96.25 | 41.01 / 91.10 / 86.33 | 48.98 / 88.30 / 96.40 |
> | JointEnergy (top-k) | 37.85 / 91.69 / 95.71 | 43.63 / 90.39 / 85.35 | 53.00 / 86.35 / 95.68 |
>
>
> We also explore how the value of k would affect the performance. As the table shows, the FPR95 decreases with the increase of k, which means more information can be aggregated. That also aligns with intuition. Note, PASCAL-VOC has 20 classes, so results for top-40 and top-60 are not applicable here.
>
> (note: we use the format of FPR95 / AUROC / AUPR in each entry below)
>
>
> | $\mathcal{D}_{\text{in}}$ | MS-COCO | PASCAL-VOC | NUS-WIDE |
> |:-------------------:|:---------------------:|:---------------------:|:---------------------:|
> | JointEnergy (top-5) | 35.83 / 92.19 / 95.99 | 41.36 / 91.00 / 86.22 | 49.87 / 87.94 / 96.28 |
> | JointEnergy (top-10) | 34.43 / 92.52 / 96.16 | 41.02 / 91.09 / 86.32 | 49.35 / 88.22 / 96.37|
> | JointEnergy (top-20) | 33.72 / 92.66 / 96.23 | 41.01 / 91.10 / 86.33 | 49.14 / 88.28 / 96.39 |
> | JointEnergy (top-40) | 33.53 / 92.70 / 96.25 | - / - / - | 48.98 / 88.30 / 96.40 |
> | JointEnergy (top-60) | 33.48 / 92.70 / 96.25 | - / - / - | 49.03 / 88.30 / 96.40 |
>
>
> **3. Ablation on log data density & joint conditional log-likelihood**
>  Another interesting suggestion! The theoretical interpretation of JointEnergy can be decomposed into two co-occurring terms. However, the exact estimation of the individual terms is not tractable. Directly estimating the joint likelihood itself can be prohibitively challenging, especially in multi-label settings as it requires sampling combinatorially in the high-dimensional space (see **L37-39**). This in fact highlights a major advantage of JointEnergy, which can be computed easily and circumvent this challenge of performing density estimation or joint likelihood estimation.
>
> Moreover, the combination of the two terms is also more advantageous than using log data density or joint conditional log-likelihood alone. This is because the addition of the two terms would increase the separability between ID and OOD data. Mathematically, let’s assume the distribution of log data density for ID data is a Gaussian with mean $\mu_1$ and OOD data with mean $\mu_2$, both with standard deviation. Since ID has higher data density, we assume $\mu_1 \ge \mu_2$ without loss of generality. Similarly, we can assume the distribution of the log-likelihood term is Gaussian with mean $\mu_3$ for ID and $\mu_4$ for OOD (and $\mu_3 \ge \mu_4$). Therefore, the combination of the two terms is a Gaussian with mean $\mu_1+\mu_3$ for ID, $\mu_2+\mu_4$ for OOD---which is more separable.
>
>
> [1] Weitang Liu, Xiaoyun Wang, John Owens, and Yixuan Li. Energy-based out-of-distribution detection. Advances in Neural Information Processing Systems, 2020.

---

### Author Response · Authors · 2021-08-10
**Summary of response -- thanks to all reviewers for thorough and insightful feedback**

 We are pleased to see that reviewers* find that the method is **novel**, **simple**, **effective**, and **powerful** (R1, R2, R3, R4), **studying a new and under-studied problem** (R1, R2, R3, R4), and the results and analysis are **thorough** (R2), **conclusive** (R3), **interesting** (R3), and **significant** (R3, R4), with **promising** and **superior performance** (R1, R2, R3, R4). We are equally glad that reviewers found the paper **well written** and **easy to follow** (R3, R4). We appreciate that R3 thinks our work can **inspire a series of works**, given its contribution to well-defining new research topics.

We have addressed the reviewers’ comments and concerns in **individual responses to each reviewer**. The reviews allowed us to improve our draft and the changes made in the revised draft are summarized below:

+ [R1] Added clarification on the connection between the training objective and the JointEnergy estimator.
+ [R1] Added discussion on top-k prediction.
+ [R1] Added discussion on log data density & joint conditional log-likelihood.
+ [R2] Added clarification on conditional energy and reference.
+ [R2] Revised the mathematical interpretation for better clarity.
+ [R3] Added discussion on limitations and future work beyond vision.
+ [R3] Added clarification on training a different model for each label.
+ [R4] Expanded the definition and property of energy in Section 3.
+ [R4] Moved clarification on the label independence assumption (footnote 1) to the main paper.


\* For brevity, we refer to reviewers **QXpR** as R1, **GeqC** as R2, **p44K** as R3, and **ggDB** as R4 respectively.

---

### Decision · Program_Chairs · 2021-09-27

**Decision:**

Accept (Poster)

**Comment:**

I recommend to accept this paper

In this paper, the authors proposed a novel method to address an important yet under explored problem: out-of-distribution detection for multi-label classification models. This paper is well-written, addressing important problem, proposing simple-but-effective method. All the reviewers are inclined to accept this paper. I would suggest the authors to add additional experimental results done in the rebuttal phase and take the reviewers' suggestions into the camera-ready version.